# EZIGEN: ENHANCING ZERO-SHOT SUBJECT-DRIVEN IMAGE GENERATION WITH PRECISE SUBJECT ENCODING AND DECOUPLED GUIDANCE

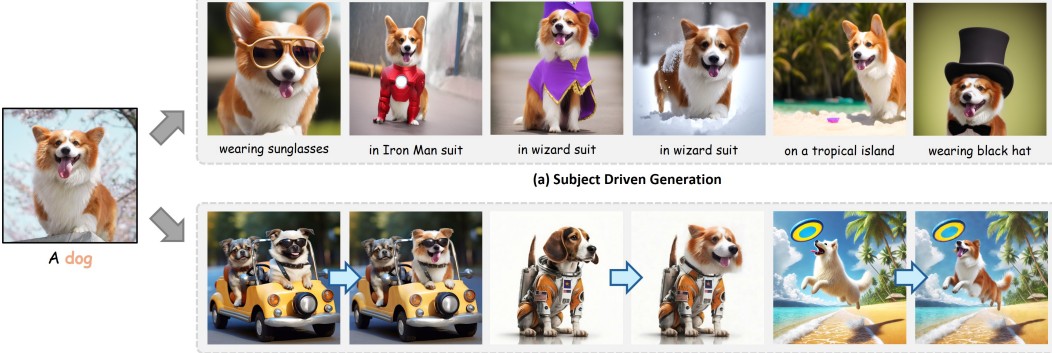

Figure 1: Our model demonstrates remarkable zero-shot performances in generating high-quality images on subject-driven generation and editing tasks.

## ABSTRACT

Zero-shot subject-driven image generation aims to produce images that incorporate a subject from a given example image. The challenge lies in preserving the subject's identity while aligning with the text prompt which often requires modifying certain aspects of the subject's appearance. Despite advancements in diffusion model based methods, existing approaches still struggle to balance identity preservation with text prompt alignment. In this study, we conducted an in-depth investigation into this issue and uncovered key insights for achieving effective identity preservation while maintaining a strong balance. Our key findings include: (1) the design of the subject image encoder significantly impacts identity preservation quality, and (2) separating text and subject guidance is crucial for both text alignment and identity preservation. Building on these insights, we introduce a new approach called EZIGen, which employs two main strategies: a carefully crafted subject image Encoder based on the pretrained UNet of the Stable Diffusion model to ensure high-quality identity transfer, following a process that decouples the guidance stages and iteratively refines the initial image layout. Through these strategies, EZIGen achieves state-of-the-art results on multiple subject-driven benchmarks with a unified model and 100 times less training data. Anonymous demo page is available at: Demo Page.

## 1 INTRODUCTION

Subject-driven generation methods enable users to create images by combining text prompts with subject images, following the principle of 'my subject' following 'my instructions.' Existing solutions fall into two categories: test-time tuning-based(Gal et al. (2022); Ruiz et al. (2023); Avrahami et al. (2023); Kumari et al. (2023); Li et al. (2024a); Hao et al. (2023)) and zero-shot inference-based(Wei et al. (2023); Ma et al. (2023); Purushwalkam et al. (2024); Chen et al. (2024)). Test-time tuning involves fine-tuning model parameters and introducing subject tokens, allowing detailed control but requiring time-consuming re-training. In contrast, zero-shot methods generate images

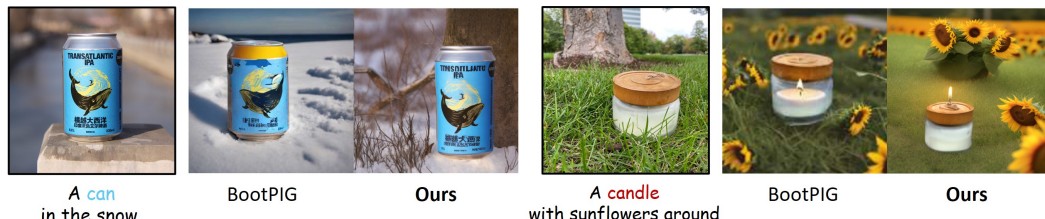

Figure 2: Suboptimal encoding. BootPIG's encoder design may lead to suboptimal performance compared to our own. "Ours" means replacing BootPIG's encoder with our encoder design.

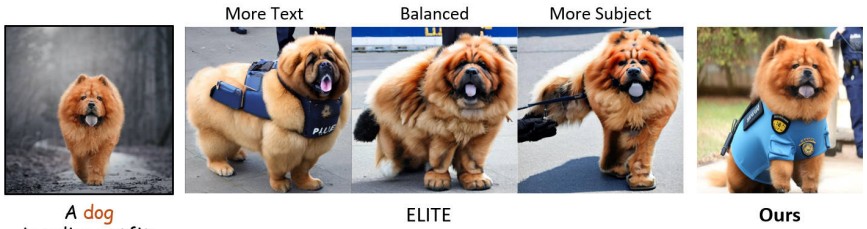

Figure 3: Conflicting guidance. Existing methods struggle to strike a good balance between identity preservation and text prompt alignment.

directly from the given subjects without re-training, offering greater efficiency. This paper focuses on improving zero-shot methods.

Existing zero-shot methods (Wei et al. (2023); Chen et al. (2024); Purushwalkam et al. (2024); Ma et al. (2023)) predominantly focus on the transfer of a reference subject's appearance into the generated image. These methods typically first encode the subject image and then integrate the encoded features into the UNet of the diffusion model. There are several existing options for subject image encoders. ELITE (Wei et al. (2023)) and Subject Diffusion (Ma et al. (2023)) utilize CLIP image encoders, while AnyDoor (Chen et al. (2024)) adopts DINOv2 to achieve better feature map extraction. BootPIG (Purushwalkam et al. (2024)) employs a Reference UNet model (Hu (2024)) as a subject feature extractor, reasoning that the feature space of a UNet would be more aligned with the feature space used in image generation. Although this approach is reasonable, it leaves many design aspects unexplored, such as which time step to use and how to inject the Reference UNet features into the generation UNet. Our research reveals that those aspects might have a big impact at the identity preservation capability.

An often overlooked challenge in existing methods is balancing the parallel user inputs: subject image and text prompt. Although these guidances appear independent—where the subject image focuses on preserving identity and the text prompt directs the model to follow user instructions—they frequently interfere with one another. For instance, as shown in Fig. 3, both the text prompt "a dog in police outfit" and the subject dog image define the dog's appearance to some extent simultaneously. In such cases, prior works either prioritize identity preservation at the expense of text coherence (Ma et al. (2023)), successfully follow the text but struggle to maintain subject identity (Purushwalkam et al. (2024)), or perform suboptimally in both aspects (Wei et al. (2023); Li et al. (2024a)).

In this paper, we present a novel subject-driven generation method, EZIGen, addressing the challenges of identity preservation and text alignment. Building on Purushwalkam et al. (2024), we employ a Reference UNet as an extractor to achieve good feature alignment between subject and generated images. However, our unique contribution lies in identifying the 'devils in the details': we discovered that using a fixed timestep, a frozen UNet, and coupling an adaptor for injecting identity information significantly enhances identity preservation compared to the approach used by BootPIG Purushwalkam et al. (2024). To better balance subject identity and text adherence, we decouple the generation process into two distinct stages: the Layout Generation Process, which forms a coarse layout from text prompts, and the Appearance Transfer Process, which injects the encoded subject details via the adapter. This decoupling explicitly separates guidance signals. Additionally, we observe that the initial layout can impact the quality of identity injection—a layout closer to the

subject tends to produce better results. Therefore, we introduce an iterative pipeline that converts the generated image back into an editable noisy latent, refining the layout with subject guidance.

With the aforementioned designs, our model delivers exceptional performance in subject identity preservation while maintaining excellent text-following capabilities in subject-driven image generation tasks, offering outstanding abilities in generating images with *various* subject poses and versatile attribute modifications. Furthermore, our model consistently performs remarkably well even in domains for which it was not specifically trained or fine-tuned. For instance, it can effortlessly generate highly detailed human facial content without any dedicated pre-training or domain-specific adjustments. Through extensive analysis, rigorous testing, and comprehensive experiments, we demonstrate that our design consistently surpasses previous methods across various benchmarks and tasks, achieving superior and reliable results with a unified, efficient approach.

## 2 RELATED WORKS

### 2.1 TEXT-TO-IMAGE GENERATION.

Generative models are designed to synthesize samples from a data distribution based on a set of training examples. These models include Generative Adversarial Networks (GANs) (Karras et al. (2021); Goodfellow et al. (2020); Brock et al. (2018)), Variational Autoencoders (VAEs) (Kingma & Welling (2013)), autoregressive models (Esser et al. (2021); Razavi et al. (2019); Tian et al. (2024); Sun et al. (2024a)), and diffusion models (Ho et al. (2020); Song et al. (2020); Dhariwal & Nichol (2021); Betker et al. (2023)). While each of these approaches has demonstrated remarkable capabilities in generating high-quality and diverse images, their inputs are typically restricted to text instructions or predefined conditions. In contrast, our work significantly enhances pre-trained diffusion models by enabling them to incorporate additional image guidance alongside text prompts, ultimately providing a more comprehensive, versatile, and flexible approach to image generation across a wider range of applications and contexts.

### 2.2 TUNNING-BASED SUBJECT-DRIVEN IMAGE GENERATION.

Tuning-based subject-driven image generation (Ruiz et al. (2023); Hao et al. (2023); Gal et al. (2022); Nam et al. (2024); Ding et al. (2024); Kumari et al. (2023); Avrahami et al. (2024); Chen et al. (2023); Liu et al. (2023)) typically adjusts sets of parameters to extend traditional text-driven methods, allowing them to incorporate additional subject images alongside text prompts, thereby enabling more personalized, detailed, and flexible image synthesis. Some approaches focus on tuning text embeddings to represent the subject accurately, such as TextualInversion (Gal et al. (2022)), which simply adjusts a learnable text embedding, and DreamBooth (Ruiz et al. (2023)), which fine-tunes both text embeddings and model parameters for more precise and effective control. ViCo (Hao et al. (2023)) and CustomDiffusion (Kumari et al. (2023)) further improve performance by additionally tuning cross-attention layers, enhancing subject appearance integration. Despite these advancements, these methods require re-training for each individual subject, making them time-consuming, and unsuitable for productivity in large-scale applications or practical deployments.

### 2.3 ZERO-SHOT SUBJECT-DRIVEN IMAGE GENERATION

To tackle the aforementioned issues, some researchers introduced zero-shot methods that accept new subjects without requiring retraining. Some attempts, such as InstantBooth (Shi et al. (2024)), FastComposer (Xiao et al. (2023)), and PhotoMaker (Li et al. (2024b)), developed zero-shot generation techniques specifically for domain-specific data, such as human faces or other constrained applications. For more general objects, ELITE (Wei et al. (2023)), BLIP-Diffusion (Li et al. (2024a)), Subject Diffusion (Ma et al. (2023)), and BootPIG (Purushwalkam et al. (2024)) achieved high-quality zero-shot generation by injecting detailed subject image features into diffusion spaces. Nevertheless, these methods still struggle with issues like degraded subject-identity preservation due to sub-optimal feature extractor utilization or poor balancing between the text and subject guidance during the generation process.

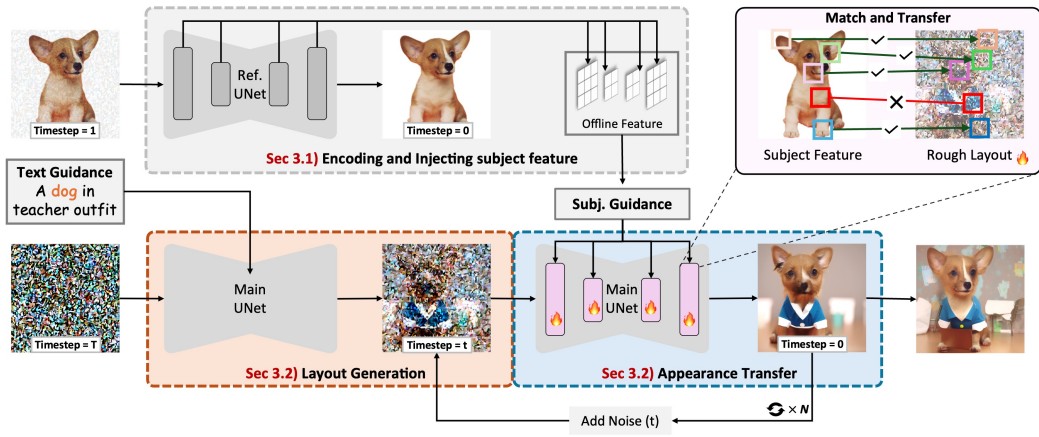

Figure 4: Illustration of the proposed system. We begin by Encoding and Injecting subject features (Sec. 3.1). Next, we decouple the generation into the Layout Generation Process and Appearance Transfer Process (Sec. 3.2). Finally, we introduce the Iterative Appearance Transfer mechanism (Sec. 3.3) to fully transfer the subject appearance feature to the layout image.

## 3 METHODS

Our method comprises two main components: a technique to encode the subject image for the generation process and a strategy to balance subject identity preservation with text alignment. We will first elaborate on these components for subject-driven image generation and then extend the discussion to subject-driven image editing.

### 3.1 ENCODING AND INJECTING SUBJECT IMAGE INFORMATION

The identity information of the subject image is extracted using an image encoder. Several options exist in current methods, such as CLIP, utilized in Subject Diffusion (Ma et al. (2023)), DINO-V2, as employed in AnyDoor (Chen et al. (2024)), and a Reference UNet initialized with the same parameters as Stable Diffusion, as seen in BootPIG (Purushwalkam et al. (2024)). The last option is particularly appealing because the noisy latent of a diffusion model and the encoding of the subject image are processed by similar models, potentially resulting in a closer feature space. This reduces the challenge of aligning the feature spaces of the subject image encoder and the diffusion UNet. However, this design leaves several open questions regarding how to configure the feature extractor and how to integrate the subject image encoding into the diffusion UNet. In BootPIG (Purushwalkam et al. (2024)), all parameters in the Reference UNet are open for fine-tuning, and the input image is progressively corrupted with noise at each timestep and then fed into the Reference UNet afterward, where the input subject image features are injected into the Main UNet at the same timestep. Our work identifies potential issues with this approach: the overly-noised feature may convey inaccurate information about the subject image, and tunning the Reference UNet would disrupt the Stable Diffusion parameters, leading to suboptimal subject encoding(Fig. 2 shows some examples). In this paper, we propose an alternative solution: inputting the image with light noise into a fixed Reference UNet to ensure accurate subject representation and introducing a learnable adapter to bridge discrepancies between latent representations from images with varying noise levels. As shown in Table 5 and Figure 9, our approach significantly improves identity preservation compared to existing methods.

As depicted in the **gray** box in Fig. 4, we derive a Reference UNet from the original Stable Diffusion model and employ it as a fixed offline subject encoder, denoted as $U_{\text{ref}}(\cdot)$. To minimize background interference, the subject image's background is removed. For extracting subject representations, we set the denoising timestep $T_{\text{sub}} = 1$ and add Gaussian noise $\epsilon$ to the subject image $x_{\text{sub}}$, depending on the $T_{\text{sub}}$ and a noise scheduler $\phi$:

$$x'_{\text{sub}} = x_{\text{sub}} + \phi(\epsilon, T_{\text{sub}}), \quad \text{where } \epsilon \sim \mathcal{N}(0, \sigma^2), \quad T_{\text{sub}} = 1 \tag{1}$$

This slightly noised image $x'_{\text{sub}}$ and the timestep $T_{\text{sub}}$ are then passed through $U_{\text{ref}}(\cdot)$ to obtain the latent representations from all $N$ self-attention layers. These representations, collectively denoted

as $\mathcal{F}_{\text{sub}}$, capture the subject-specific features:

$$\mathcal{F}_{\text{sub}} = \{s_1, s_2, \ldots, s_N\} = U_{\text{ref}}(x'_{\text{sub}}, T_{\text{sub}}) \tag{2}$$

To integrate these subject features, we introduce an Adapter as an additional attention module situated between the self-attention and cross-attention blocks within each transformer block of the Main UNet, resulting in a total of $N$ Adapters. For each Adapter $A_n$, the Main UNet's latent feature $x'$ is projected into query, key, and value matrices $Q$, $K$, and $V$, respectively. Meanwhile, the subject feature $s_n$ is mapped into additional key $K_{\text{sub}}$ and value $V_{\text{sub}}$ matrices as follows:

$$K_{\text{sub}} = W_k s_n, \quad V_{\text{sub}} = W_v s_n \tag{3}$$

The output feature $\mathcal{F}_n$ for each Adapter $A_n$ is then computed by combining the Main UNet's features with the subject features through the following attention operation:

$$\mathcal{F}_n = A_n(Q, [K; K_{\text{sub}}], [V; V_{\text{sub}}]) \tag{4}$$

Here, we have $\forall n \in \{1, 2, \ldots, N\}$, $[K; K_{\text{sub}}]$ and $[V; V_{\text{sub}}]$ represent the concatenation of feature vector $K$ with $K_{\text{sub}}$ and $V$ with $V_{\text{sub}}$ along the token dimension.

**Training the Adapter.** We follow standard practices in the field to construct subject image pairs from image/video datasets as training data. In each pair, one image serves as the subject guidance, while the other is treated as the target. During training, all parts of the model are fixed except for the Adapter, which learns to recover the noisy target image under the guidance of the subject features.

## 3.2 DECOUPLING TEXT AND SUBJECT GUIDANCE

As mentioned previously, existing methods (Wei et al. (2023); Ma et al. (2023); Purushwalkam et al. (2024)) often struggle to balance subject ID preservation, i.e. subject guidance, with text adherence, i.e. text guidance. While the design in Sec. 3.1 excels in preserving subject identity, it still faces challenges in achieving this balance. We observe that injecting subject features alongside text prompts at all timesteps tends to prioritize subject identity, overshadowing text-guided semantic layouts and color patterns that are not fully established in the early stages, such as the "teacher's outfit." Instead of using parallel guidance with scaling factors (Wei et al. (2023); Purushwalkam et al. (2024); Ma et al. (2023)), which often compromises one aspect, we decouple the guidances to let them dominate at different stages: text guidance in the early stages and subject guidance details later. This leads to two distinct sub-processes: the Layout Generation and Appearance Transfer Process.

**Layout Generation Process.** First, we take the original text prompt as text guidance and generate a coarse layout using the Stable Diffusion model. Specifically, we interrupt the generation process at a certain timestep $T_{\text{layout}}$ and regard the intermediate latent as the coarse layout latent, denoted as $x'$, containing the overall semantic structure and rough color patterns of the image, as shown by the last image of the orange box in Fig. 4.

**Appearance Transfer Process.** Then, as shown in the blue box in Fig. 4, we bring in subject feature $\mathcal{F}_{\text{sub}}$ as subject guidance and transfer the subject appearance to the layout $x'$ using the adapters $A$. Intuitively, we discover that the attention mechanism within the adaptor first establishes matching (represented by the attention map) between subject image patches from the Reference UNet and the noisy latents from the Main generation UNet, which define the scene's initial layout. It then transfers the content (encoded by the V values) from the subject patches to their corresponding locations in the image being generated. This understanding can be illustrated in Fig. 4, given a rough layout from the Layout Generation Process depicting "a dog in a teacher outfit", our model will first match between the subject dog and the dog in the rough latent layout, then transfer only the paired brown furry skin and maintain the blue teacher's outfit untouched.

## 3.3 ITERATIVE APPEARANCE TRANSFER

Based on the above analysis and empirical results, we observe that the initial layout can influence the final subject-driven generation outcomes. When the initial layout resembles the subject image, the transfer process improves as the adapter establishes better correspondence within the Main UNet's noisy latent space. To enhance this, we introduce an iterative generation scheme: after each transfer, the generated image becomes the new noisy layout, with noise added according to timestep $T_{\text{layout}}$

for further editing, as shown in the bottom part of Fig. 4. This process repeats until the similarity between the newly generated image and the previous image exceeds a predefined threshold, indicating that the appearance transfer is complete and no further information is added.

Integrating the designs above, our model successfully balances subject identity preservation and text adherance, ensuring comprehensive guidance-following without compromise.

### 3.4 SUBJECT-DRIVEN IMAGE EDITING

We discover that the Appearance Transfer Process can naturally function as an effective subject-driven image editor. This is achieved by integrating an object mask and replacing noise addition with image inversion (Lu et al. (2023)), which converts the generated image back into the latent layout, preserving the background. Specifically, similar to the noise addition described in Sec. 3.3, given a real image, we first partially invert it based on timestep $T_{\text{layout}}$ to obtain a coarse layout latent $x'_{\text{r}}$. Next, we initiate the iterative appearance transfer process to inject the subject feature, resulting in an incomplete edited latent $\hat{x}'_{\text{r}}$. To maintain the original background, we separate the edited foreground from $\hat{x}'_{\text{r}}$ and the background from $x'_{\text{r}}$ using a user-provided foreground mask $M$:

$$x_r^{\text{comb}} = M \otimes \hat{x}'_{\text{r}} + (1 - M) \otimes x'_{\text{r}} \tag{5}$$

Here, $x_r^{\text{comb}}$ combines both the static background and the desired edition result in the foreground, we can then use $x_r^{\text{comb}}$ as the starting point for the next iteration.

## 4 EXPERIMENT

### 4.1 IMPLEMENTATION DETAILS

**Benchmark and Evaluation**. We evaluate our design on three benchmarks: DreamBench (Ruiz et al. (2023)) for subject-driven image generation, DreamEdit (Li et al. (2023)) for subject-driven image editing, and FastComposerBench (Xiao et al. (2023)) for human content generation. For DreamBench and DreamEdit, we follow Subject Diffusion's protocol, averaging scores over 6 random runs. For human content generation, we use the FastComposer API. For DreamBench, we report CLIP-T, CLIP-I, and DINO scores to assess text adherence and subject identity. For subject-driven editing, we evaluate DINO/CLIP-sub for foreground similarities and DINO/CLIP-back for background, using SAM (Kirillov et al. (2023)) for mask extraction. For the aforementioned tasks, we take the DINO-related scores as the main metric for subject similarity, as it better depicts the detailed patch-level differences between images. Finally, for human content generation, we calculate ID preservation and prompt consistency, following Xiao et al. (2023).

**Training Dataset Construction**. We create training pairs from COCO2014 and YoutubeVIS. In COCO2014, 1-4 objects are cropped from target image as subject images, while in YoutubeVIS, we extract images of the same subject from different frames as pairs, following Chen et al. (2024). We sampled 100k pairs from each dataset, totaling 200k pairs.

**Experiment settings**. We utilize Stable Diffusion V2.1-base, with the image resolution fixed at 512×512 for all experiments. The adapter is initialized from the self-attention module within each transformer block to enhance compatibility. We train the model for 1 epoch using a batch size of 1, with the Adam optimizer and a learning rate of $1e-5$. During inference, iterations are designed to stop automatically when the newly generated image exhibits a sufficiently high similarity with the image from the previous loop, ensuring efficient convergence and maintaining generation quality.

### 4.2 COMPARING WITH EXISTING SUBJECT-DRIVEN GENERATION METHODS

In Tab. 1, we compare our method with existing zero-shot subject-driven approaches on the Dream-Bench dataset (Ruiz et al. (2023)), including tuning-based methods for reference. We also present the number of reference images required for each subject and the training dataset size to highlight the cost of each method. The overall results show that our base model achieves state-of-the-art performance text-following, subject identity preservation, and balance among all methods. Compared to the previous state-of-the-art, Subject Diffusion (Ma et al. (2023)), our model outperforms in both CLIP-T (0.316 *vs.* 0.293) and DINO score (0.718 *vs.* 0.711), demonstrating superior flexibility in generated images and higher subject identity preservation, without sacrificing balance.

| Method | CLIP-T | CLIP-I | DINO | # Sub | TS | DS | ZS |
|---|---|---|---|---|---|---|---|
| Textual Inversion ‡ | 0.261 | 0.772 | 0.561 | 3-6 | N/A | N/A | ✗ |
| DreamBooth ‡ | 0.306 | 0.792 | 0.672 | 4-6 | N/A | N/A | ✗ |
| Elite ‡ | 0.296 | 0.772 | 0.647 | 1 | multi | **0.125M** | ✓ |
| BLIP-Diffusion ‡ | 0.298 | 0.779 | 0.589 | 1 | multi | ∼2M | ✓ |
| BootPIG (4-6 ref) † | 0.311 | **0.797** | 0.674 | 4-6 | single | 0.2M | ✓ |
| Subject Diffusion † | 0.293 | 0.789 | 0.711 | 1 | single | ∼76M | ✓ |
| **Ours** | **0.316** | 0.782 | **0.718** | **1** | single | 0.2M | ✓ |

Table 1: Quantitative comparison on DreamBooth benchmark. "# Sub" means the number of images of the same subject required for training and inference, "TS" for training stages, "DS" for Dataset Size, and "ZS" for zero-shot inference. ‡ indicates that the results are taken from the Subject Diffusion (Ma et al. (2023)) paper, † are from the original paper.

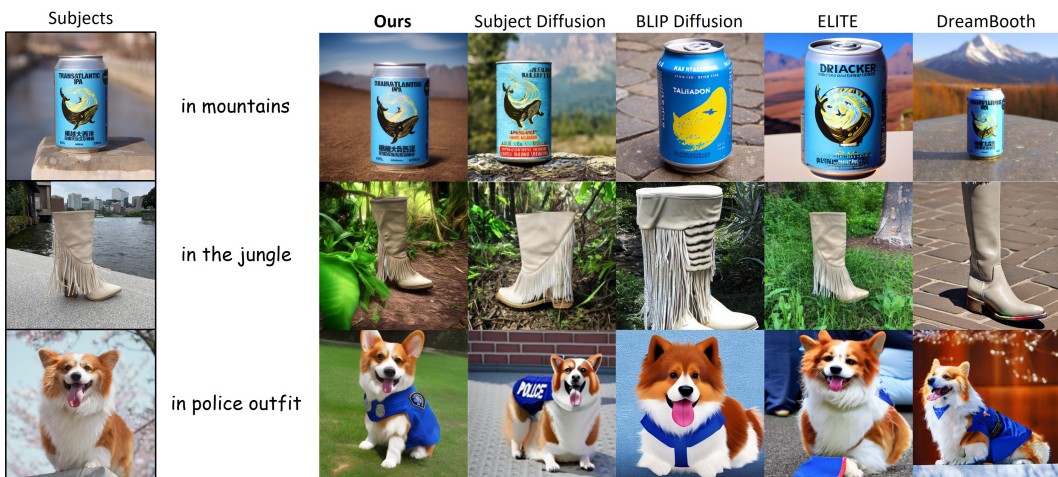

Figure 5: Comparison with existing subject-driven generation methods. Since Subject Diffusion is not publically available, we take resultant images from the original paper.

Moreover, this was accomplished with 100× less training data, which we attribute to our simplified design and the use of the Reference UNet. These results highlight the advantages of using a Reference UNet over CLIP for feature extraction, alongside our decoupling design that improves text coherence. When compared with BootPIG (Purushwalkam et al. (2024)), our method surpasses it in text-following ability (0.316 *vs.* 0.311) and significantly outperforms in DINO score (0.718 *vs.* 0.674), while requiring fewer subject images. This is due to our more advanced utilization of the Reference UNet. Additionally, our method outperforms previous open-source zero-shot methods (Wei et al. (2023); Li et al. (2024a)) across all evaluation metrics.

### 4.3 VALIDATION ON SUBJECT-DRIVEN EDITING TASK

As outlined in Sec. 3.4, our method adapts well to subject-driven image editing by replacing noise addition with image inversion (Lu et al. (2023)) in the Appearance Transfer Process. We compare our model against previous state-of-the-art methods on the DreamEdit benchmark, as shown in Tab. 2 and Fig. 6(1). Using DINO-sub as the main metric for subject fidelity, our method significantly outperforms others, primarily due to our advanced subject feature utilization, where the Reference UNet extracts high-quality, detailed features, and the Adapter accurately matches subject appearances to the layout. Additionally, our approach achieves superior background preservation, thanks to the explicit separation of foreground and background, which effectively confines inversion errors and pixel alterations strictly to the foreground, leaving the background entirely intact.

---

[1] We follow DreamEdit to take the result images from ImageHub evaluation platform.

| Method | DINO sub | DINO back | CLIP-I sub | CLIP-I back |
|---|---|---|---|---|
| DreamBooth‡ | 0.640 | 0.427 | **0.811** | 0.736 |
| PhotoSwap‡ | 0.494 | **0.797** | 0.751 | **0.889** |
| DreamEdit‡ | 0.627 | 0.574 | 0.784 | 0.821 |
| **Ours** | **0.650** | 0.792 | 0.782 | **0.889** |

Table 2: Scores on DreamEditBench. The scores show the effectiveness of our subject encoding. Results with ‡ are referenced from DreamEdit.

| Method | ID Preser. | Prompt Consis. |
|---|---|---|
| DreamBooth‡ | 0.273 | 0.239 |
| FastComposer‡ | 0.514 | **0.243** |
| SubjectDiffusion‡ | **0.605** | 0.228 |
| **Ours** | 0.592 | 0.236 |

Table 3: Performances on single-subject human image generation. Results with ‡ are referenced from Subject Diffusion.

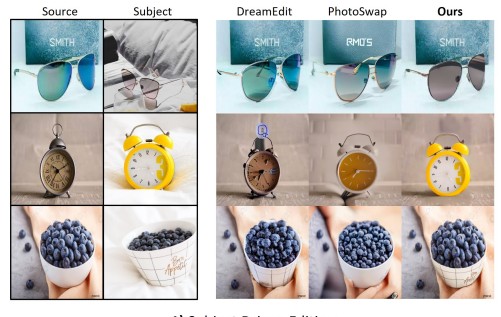

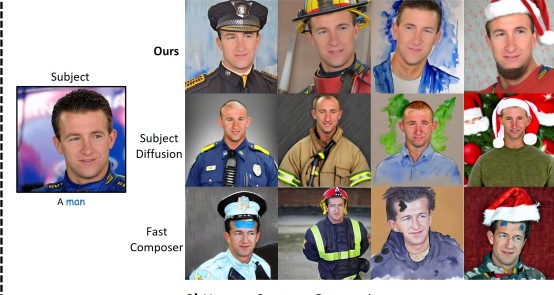

1) Subject Driven Editing  2) Human Content Generation

Figure 6: Comparing on 1) subject-driven editing[1] and 2) human content generation tasks.

## 4.4 EVALUATING PERFORMANCES ON HUMAN CONTENT GENERATION TASK.

We demonstrate the effectiveness of our method for subject-driven human image generation. Unlike FastComposer (Xiao et al. (2023)), which relies on domain-specific datasets, or Subject Diffusion (Ma et al. (2023)), which requires large-scale pretraining, our model achieves high-quality results using only a normal-scale open-domain dataset. As shown in Tab. 3, our approach delivers very competitive results. In terms of subject ID preservation, we match Subject Diffusion and outperform FastComposer. While our prompt consistency lags behind FastComposer and DreamBooth, it still exceeds Subject Diffusion. These findings are also reflected in Fig. 6(2), where our method accurately captures facial details, such as shape, hair, and facial landmarks.

## 4.5 ABLATION STUDY

We conducted ablation studies to evaluate the effects of various components in our method, including the Subject Encoding and Injection module, the naive single-stage Appearance Transfer, and the Iterative Appearance Transfer process. The results are reported correspondingly in Tab. 4 and Fig. 7, with markings from 1) to 4).

**Effectiveness of UNet-based Subject Encoding and Injection.** To evaluate the quality of the encoded subject feature, we start with a fully noised layout image (or random noise) and apply subject guidance at all timesteps. As shown in experiments 1) and 2), compared to the original text-guided generation, the encoded subject representation demonstrates strong identity preservation, achieving a DINO score of 0.762 and a CLIP-I score of 0.808. However, the strong subject features tend to override the weaker layout latent in early timesteps, leading to serve copy-paste effect and a lower CLIP-T score of 0.286.

**Decoupling the generation process.** The experiment in 3) confirms the effectiveness of separating the Layout Generation Process from Appearance Transfer, in which the CLIP-T score improves significantly from 0.286 to 0.321, as this design explicitly maintains the layout. However, when there is a large discrepancy between the layout content and the subject, the appearance transfer can be incomplete, resulting in reduced subject similarity scores compared to experiment 1), where the DINO score drops from 0.762 to 0.560.

**Effectiveness of Iterative Appearance Transfer.** Experiment 4) indicates a successful appearance transfer after introducing the iterative process. Such gentle refinement to the incomplete generated

| #ID | Encode & Inject | Decoupled Generation Process | Iterative App. Transfer | CLIP-T | CLIP-I | DINO |
|---|---|---|---|---|---|---|
| 1 | | | | 0.327 | 0.658 | 0.362 |
| 2 | ✓ | | | 0.286 | **0.808** | **0.762** |
| 3 | ✓ | ✓ | | 0.321 | 0.714 | 0.560 |
| 4 | ✓ | ✓ | ✓ | **0.316** | 0.782 | 0.718 |

Table 4: Quatitative ablation study on DreamBench dataset.

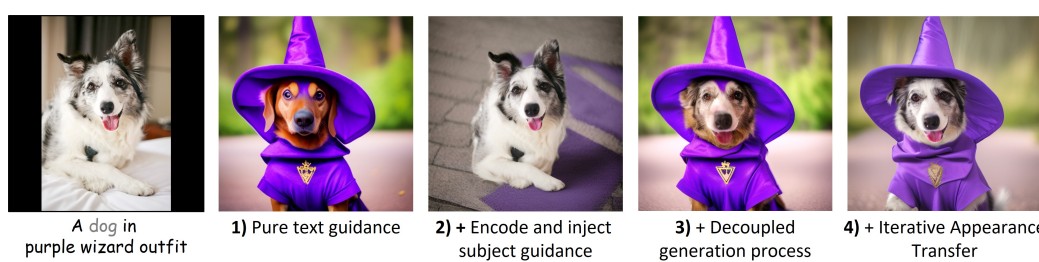

A dog in purple wizard outfit    **1)** Pure text guidance    **2) +** Encode and inject subject guidance    **3) +** Decoupled generation process    **4) +** Iterative Appearance Transfer

Figure 7: Qualitative result of the ablation study.

image leads to substantial improvements in subject appearance similarity: CLIP-I improved from 0.714 to 0.782, DINO score from 0.560 to 0.718, while the text-guided semantics of the generation images are slightly disrupted during the iterative process, and thus the CLIP-T score drop marginally from 0.321 to 0.316, our model reaches the best balance between two guidance under this setting.

### 4.6 SUPERIORITY OF OUR REFERENCE UNET DESIGN AMONG OTHER ALTERNATIVES

We evaluate the effectiveness and efficiency of our Reference UNet subject extractor compared to CLIP, DINO, and alternative Reference UNet configurations. We use three baseline methods—Subject Diffusion (Ma et al. (2023)), AnyDoor (Chen et al. (2024)), and BootPIG (Purushwalkam et al. (2024))—as the current best practices for feature extraction. To ensure a fair comparison, we ablate the models for AnyDoor and Subject Diffusion (lines 2 and 4) so that subject information is derived solely from the feature extractor.

**Comparing with CLIP- and DINO-based methods.** First, to evaluate the efficiency of using Reference UNet versus CLIP and DINO, we directly replaced our "Reference UNet + Adapter" combination with "CLIP/DINO + Projector + Adapter" and compared performances under identical training settings. As shown in Tab. 5 lines (a) and (c), this replacement led to trivial solutions, as the Adapter struggled to establish attention between misaligned feature maps, causing the model to rely primarily on the text prompt for reconstruction. To address this, additional regularization is typically required to overcome feature space misalignment and force the model to follow the subject guidance. For example, Subject Diffusion employs location control as regularization, while AnyDoor replaces text tokens with image tokens and crops a scene image for the subject feature to fill. However, even with regularization, the required dataset sizes remain substantial, at approximately 76M and 9M image pairs, respectively. In contrast, with closer feature spaces, our method completes training with only 0.2M image pairs. Second, to validate the effectiveness of our design over CLIP and DINO extractors in encoding subject information, we report the DINO and CLIP-I scores on the DreamBench dataset using the same evaluation protocol, as shown in Tab. 5 lines (b) and (d), and Fig. 8, our Reference UNet significantly outperforms both DINO- and CLIP-based configurations. This suggests that, although CLIP and DINO are highly effective for high-fidelity image representation across various downstream tasks (Liu et al. (2024); Zhou et al. (2023); Sun et al. (2024b)), the Stable Diffusion UNet is more suitable for providing features for high-quality image generation, as it was specifically designed.

**Evaluating alternative Reference UNet configurations.** We examined the impact of tuning Reference UNet parameters and adding varying levels of noise to subject images, as shown in Fig. 9. The table on the left lists the experiment settings and their performances, where check marks/cross marks indicate whether the Reference UNet is trained or frozen. "Adaptive" means that the Reference UNet shares the same denoising timestep as the Main UNet, while arguments starting with "$T_{sub}$" indicate the level of noise we add to the subject image. From lines (a) and (b), we find that

| Feature Extractor | Training Strategies | CLIP-I | DINO | Dataset Size |
|---|---|---|---|---|
| DINO | a) DINO + Projector | 0.661 | 0.421 | 0.2M |
| | b) AnyDoor *w/o high-freq map* † | 0.775 | 0.710 | ∼9M |
| CLIP | c) CLIP + Projector | 0.682 | 0.433 | 0.2M |
| | d) Subject Diff. *w/o image CLS token* † | 0.719 | 0.637 | ∼76M |
| Fixed Ref UNet | e) Ours *w/ all-step transfer* | **0.808** | **0.762** | **0.2M** |

Table 5: Comparing extractor designs. Results with † are taken from the original paper.

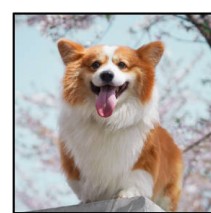 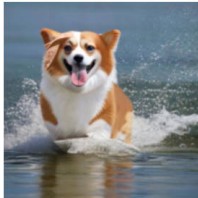 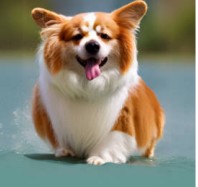 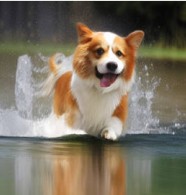

A dog running on water.    Subject Diffusion    AnyDoor    **Ours**

Figure 8: Comparison with state-of-the-art CLIP-based and DINO-based methods Subject Diffusion (Ma et al. (2023)) and AnyDoor (Chen et al. (2024)).

| #ID | Train Ref. UNet | Ref. UNet Timestep | CLIP-I | DINO |
|---|---|---|---|---|
| a) BootPIG* | ✓ | Adaptive | 0.764 | 0.589 |
| b) | ✗ | Adaptive | 0.771 | 0.615 |
| c) | ✓ | $T_{sub}=1$ | 0.778 | 0.668 |
| d) | ✗ | $T_{sub}=800$ | 0.427 | 0.427 |
| e) | ✗ | $T_{sub}=500$ | 0.679 | 0.598 |
| f) | ✗ | $T_{sub}=300$ | 0.789 | 0.661 |
| g) | ✗ | $T_{sub}=100$ | 0.789 | 0.725 |
| h) Ours | ✗ | $T_{sub}=1$ | **0.808** | **0.762** |

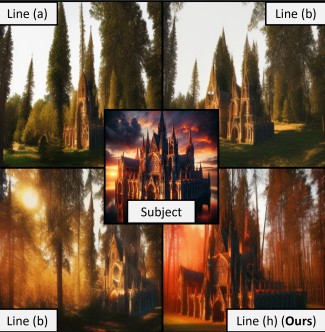

Figure 9: Evalutating subject ID preservation under different UNet configurations. We obtain the best result when we fix the Reference UNet parameters and assign it with a small denoising timestep $T_{sub}$. "BootPIG*" indicates BootPIG's training strategy.

tuning the Reference UNet disrupts the original denoising capabilities of Stable Diffusion, resulting in a performance drop compared to freezing it. Lines (a) and (c) show that fixing the denoising timestep of the Reference UNet to a small $T_{sub}$ produces better subject identity with clearer feature maps. By fixing the Reference UNet and adjusting $T_{sub}$, we validated its importance; as shown in lines (d) to (h), subject identity scores increase as $T_{sub}$ decreases. This aligns with the intuition that the Reference UNet encodes clearer features in the final timesteps of the generation process.

## 5 CONCLUSION

In this work, we introduced EZIGen, a novel framework for zero-shot subject-driven image generation. By adopting a carefully designed Reference UNet from the Stable Diffusion model, our method excels in subject feature extraction, allowing for superior subject identity preservation. Then, by explicitly separating text and subject guidances and proposing the Iterative Appearance Transfer Process, we demonstrated how our approach balances identity preservation with text-prompt coherence, surpassing the limitations of prior methods that often struggle to achieve this balance. With extensive experiments across multiple benchmarks and on inner-model analysis, our model demonstrates its ability to serve as a robust and versatile solution for subject-driven image generation tasks.

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
