# OpenReview forum: "EZIGen: Enhancing zero-shot subject-driven image generation with precise subject encoding and decoupled guidance"
_ICLR.cc/2025/Conference — ICLR 2025 Conference Withdrawn Submission_

### Official Review · Reviewer_Pgkk · 2024-10-24

**Soundness:** 2
**Presentation:** 2
**Contribution:** 2
**Rating:** 5
**Confidence:** 4

**Summary:**

This paper tackles a challenging and attractive topic, identity preservation. This paper proposes a new approach named EZIGen, which encode and inject subject feature and decouple the generation process into layout generation and appearance process, followed by an iterative appearance transfer mechanism to fuse the subject appearance to the given layout. A series of experiments are provided to demonstrate its effectiveness.

**Strengths:**

1. The method section is clear and easy to understand.
2. The experiment part contains abundant experiments and detailed ablations are provided for analysis.
3. Abundant details are provided in the paper, which could contribute to the reproduction of this work.

**Weaknesses:**

1. Novelty: The idea of encoding features from given images and injecting them into the Unet has become a very common method for current identity preservation methods. Besides, the idea of adding features from pre-trained UNET is somewhat like the famous ControlNet method. The difference is that ControlNet fine-tunes the controlnet work while this work fine-tunes the main unet.
2. The method is complex and requires three UNET in the generation process (ref. unet, main unet (fixed) and main unet (trainable)), which would limit the application scenarios with very large computational cost and resources.
3. Quantitative Performance:  This paper provides several quantitative metrics to demonstrate its effectiveness. However, the proposed method fails to achieve SOTA results on many metrics and also fails to make significant improvements on other metrics. It makes me doubt about the contribution made by this work.
4. Qualitative Performance: This paper provides many visualized examples. However, I find most of them are very simple, the subjects are simple and the layout are basically in the center of images. For example, the clock in Fig. 6 1) is just the clock example shown in DreamBooth. So I wonder why prior works cannot deal with it correctly. Besides, there exist beard chaneg in the example of Fig. 6 2) as well.
5. In addition, for the example of subject interpolation of a dog astronaut, I found quality degradation of other parts and unnatural animal heads (with blurs and artifacts).

**Questions:**

1. I wonder which one would be better? Fine-tuning the mian unet v.s. fine-tuning the ref. unet like ControlNet.
2. I find several limitations in the experiments section. See details in weakness.
3. This paper is implemented with SD2.1, what about other modern models, e.g., SDXL, SD3 medium?

---

### Official Review · Reviewer_s6XJ · 2024-10-25

**Soundness:** 2
**Presentation:** 2
**Contribution:** 1
**Rating:** 5
**Confidence:** 4

**Summary:**

This paper introduces EZIGen, a method for zero-shot subject-driven image generation that aims to enhance identity preservation while maintaining alignment with textual prompts. It proposes separating text and subject guidance while training a specific adaptor with a fixed UNet to transfer appearance. The method is evaluated on both subject generation and editing tasks.

**Strengths:**

1. The paper provides valuable insights into the challenges of balancing text prompts with subject identity in zero-shot image generation. By identifying how conflicting guidance signals often disrupt subject alignment, the authors propose a decoupling method that separates text-driven layout generation from subject appearance transfer.

2. The proposed EZIGen model is rigorously evaluated on widely used benchmarks, including DreamBench and DreamEdit, which assess both subject-driven generation and editing tasks.

**Weaknesses:**

1. The method’s use of distinct prompts at different time steps, while beneficial, has been thoroughly explored in prior work, notably in methods such as “P+: Extended Textual Conditioning in Text-to-Image Generation.” This limits the novelty of the contribution in EZIGen, as previous research already demonstrates the benefits of staging prompts to improve detail and fidelity in generated images

2. The experiments rely heavily on quantitative benchmarks without incorporating human assessments, which are crucial for assessing subjective factors like image realism, diversity, and prompt alignment in a nuanced manner.

3. While the paper identifies a crucial problem—differences between the layout of the generated image and the reference subject image—it proposes maintaining layout similarity as a solution. This approach, however, may inadvertently limit text-prompt alignment, especially when prompts require transformations or settings different from the original subject image. Addressing this conflict with a more adaptive mechanism could improve alignment without constraining diversity.

**Questions:**

See weaknesses.

---

### Official Review · Reviewer_7uus · 2024-10-30

**Soundness:** 3
**Presentation:** 3
**Contribution:** 3
**Rating:** 6
**Confidence:** 5

**Summary:**

The paper introduces a novel approach, EZIGen, to zero-shot subject-driven image generation. It addresses the challenge of balancing between preserving the identity of a subject and aligning it with a text prompt. EZIGen has a subject image encoder based on the pretrained UNet of the Stable Diffusion model and a decoupled guidance strategy that separates text and subject guidance into different stages. It proposed an Iterative generation framwork that allows the model to refine the result by iterations. The overall approach allows EZIGen to achieve state-of-the-art results on multiple benchmarks with significantly less training data compared to existing methods.

**Strengths:**

1. The proposed iterative generation scheme is an intriguing framework that has the potential to inspire a new paradigm in the field of image generation and editing. In the domain of Large Language Models (LLMs), test-time computation is becoming a prevalent paradigm. A similar philosophy could also be beneficial to image generation, as demonstrated in this paper.
2. The problem addressed by this method is of high value. Balancing the guidance of image prompts and text prompts in image generation is a challenging issue in existing methods, and this approach effectively addresses this challenge.
3. Besides the iterative generation scheme, the other parts of the paper are logically sound, and the experimental results indicate a significant improvement in generation quality.

**Weaknesses:**

1. The comparison with existing methods is somewhat lacking, particularly with methods such as [1, 2, 3] that focus on image prompt adaptation and subject-driven generation.
2. The experiments do not include a user study.
3. The strategy of "repeating until the similarity between the newly generated image and the previous image exceeds a predefined threshold" may lead to overconsumption. In my understanding, the iteration number possibly exceeds 10 if the threshold is wrongly set.
4. There is a lack of analysis on how performance changes with an increase in the number of iteration steps.

[1] IP-Adapter: Text Compatible Image Prompt Adapter for Text-to-Image Diffusion Models
[2] SSR-Encoder: Encoding Selective Subject Representation for Subject-Driven Generation
[3] MS-Diffusion: Multi-subject Zero-shot Image Personalization with Layout Guidance

**Questions:**

1. The use of the term "layout" in the paper is somewhat ambiguous. There are existing layout-based methods for image generation, such as [4] and [5], where the layout is estimated to guide the image generation process. This form of layout is more common and differs from its use in this paper. Therefore, I suggest the authors consider using a different term to avoid confusion.
2. How is the matching achieved in the "Match and Transfer" block in Figure 4? There is no description of how point matching is obtained in the manuscript.
3. What is the average number of iterations used in the experiments? Can this number be reduced?

[4] Omost
[5] Mastering text-to-image diffusion: Recaptioning, planning, and generating with multimodal LLMs

---

### Official Review · Reviewer_D2UN · 2024-11-04

**Soundness:** 3
**Presentation:** 2
**Contribution:** 2
**Rating:** 5
**Confidence:** 4

**Summary:**

The paper presents a zero-shot subject-driven image generation framework. Identity preservation and text prompt coherence are balanced using a proposed iterative appearance transfer process. Experiments on 3 benchmarks demonstrate the effectiveness of the proposal.

**Strengths:**

The paper is clearly written and easy to follow. Experiments on multiple datasets are sufficient to demonstrate the effectiveness of the proposal in subject-driven generation. The appearance transfer process helps to maintain the texture of the reference image.

**Weaknesses:**

There are several issues limiting the quality of the paper. First, extracting features from the reference image and injecting it to the generation process for identity preservations is a widely used stratagem. Second, the proposed method decouples the generation process into three parts, namely injection, layout generation, and appearance transfer. This may lead to extra complexity compared to other methods. Actually, it seems that the appearance transfer step contributes most to the visual identity preservation effect. Such a process can actually be added to any existing method to improve visual quality. Third, according to Tables 1,2,3, the performance improvements of the proposal is limited. Also, some of the visual results are not convincing enough. Also, it seems to me that the diversity of the object appearance is too limited probably due to the appearance transfer module. For example, in Figure 6 2), it seems that the face of the subject is simply pasted to different backgrounds.

**Questions:**

1.	As pointed out in the weakness part, how to balance the identity preservations and generation diversity?
2.	What is the complexity of the proposal compared to previous methods?
3.	According to Figure 4, the proposed model is trained during inference. If so, are the comparisons fair?

---

### Note · Authors · 2024-11-15

I have read and agree with the venue's withdrawal policy on behalf of myself and my co-authors.